# Adipose Tissue as a Major Launch Spot for Circulating Extracellular Vesicle-Carried MicroRNAs Coordinating Tissue and Systemic Metabolism

**DOI:** 10.3390/ijms252413488

**Published:** 2024-12-17

**Authors:** Paula Diez-Roda, Elena Perez-Navarro, Ruben Garcia-Martin

**Affiliations:** Department of Immunology and Oncology, Centro Nacional de Biotecnología (CNB)-CSIC, 28049 Madrid, Spain; paula.diez@cnb.csic.es (P.D.-R.); elena.perez@cnb.csic.es (E.P.-N.)

**Keywords:** adipose tissue, microRNAs, extracellular vesicles, exosomes, metabolism, intercellular communication

## Abstract

Circulating microRNAs (miRNAs), especially transported by extracellular vesicles (EVs), have recently emerged as major new participants in interorgan communication, playing an important role in the metabolic coordination of our tissues. Among these, adipose tissue displays an extraordinary ability to secrete a vast list of EV-carried miRNAs into the circulation, representing new hormone-like factors. Despite the limitations of current methodologies for the unequivocal identification of the origin and destination of EV-carried miRNAs in vivo, recent investigations clearly support the important regulatory role of adipose-derived circulating miRNAs in shaping the metabolism and function of other tissues including the liver, muscle, endocrine pancreas, cardiovascular system, gastrointestinal tract, and brain. Here, we review the most recent findings regarding miRNAs transported by adipose-derived EVs (AdEVs) targeting other major metabolic organs and the implications of this dialog for physiology and pathology. We also review here the current and potential future diagnostic and therapeutic applications of AdEV-carried miRNAs.

## 1. Introduction

Adipose tissue was traditionally observed as a long-term energy storage organ, but it is now recognized as a tissue playing a major role in the integration of systemic metabolism. This is mediated, at least in part, by its ability to secrete numerous protein factors, named adipokines [1]. These proteins have diverse regulatory functions both locally and in other tissues on insulin sensitivity, inflammation, energy balance, and many other processes [1,2]. However, recent research has greatly expanded the repertoire of non-protein molecules that participate in intercellular communication, representing novel adipokine-like messengers. These include several lipids (termed lipokines [3]) and RNAs, especially miRNAs. miRNAs are short non-coding single-stranded RNAs produced by virtually all cells in our body that play a pervasive role in posttranscriptional regulation, estimated to control the expression of up to 60% of our genes [4]. Transcribed as double-stranded pri-miRNAs, they are later processed by Drosha and Dicer to generate a miRNA duplex, from which one or the two strands constitute the mature miRNAs that interact with Argonaute (AGO) [4]. AGO constitutes an integral part of the miRNA-Induced Silencing Complex (RISC) that ultimately leads to the repression of mRNA targets mainly by two complementary molecular mechanisms: mRNA destabilization/decay and translational repression. Briefly, once AGO is loaded with a single-stranded miRNA, it scans the transcriptome, searching for complementary mRNA targets. In most cases, miRNA only pairs through its first 2–8 nucleotides from the 5’ end (called the seed region) with the mRNA target, often within the 3′-untranslated region (3′UTR) [4,5]. This leads to the recruitment of the adaptor protein TNRC6, the polyA-binding protein PABPC, and several deadenylases (e.g., CNOT, PAN2-PAN3, etc.), which shorten the polyA-tail of the target mRNA [6,7]. In addition, recruitment of DDX6 and 4E-T permits the cap-binding protein 4EHP to displace eIF4E from the 5′cap, leading to mRNA decapping. Both processes (3′polyA deadenylation and 5′ decapping) result in mRNA destabilization and decay through the action of exonucleases [6,7]. In addition, the dissociation of eIF4E from the cap structure impedes ribosomal assembly and scanning of the target mRNA, thus leading to translational repression [6,7]. Moreover, in rare cases in mammals where miRNA and target mRNA extensively pair beyond the seed region, AGO (mainly AGO2) directly slices the mRNA target, resulting in stronger repression [4]. The latter is the predominant mode of action of small interfering RNAs (siRNAs). siRNAs are short RNAs similar in size to miRNAs that, by directly incorporating into the RISC complex, also lead to the inhibition of target genes. However, siRNAs often target a unique gene by full complementarity whereas miRNAs typically exert global inhibitory effects on a given biological process by targeting multiple genes within the same or related pathway.

Due to the abundance of RNases in blood and other extracellular biofluids [8], the possibility of RNAs (including miRNAs) participating in long-distance intercellular communication was never deemed as possible until the discovery of extracellular miRNA carriers. The first identifications of miRNAs as novel participants in intercellular communication were made in the late 2000s, when they were found to be carried by extracellular vesicles (EVs) and delivered to recipient cells, leading to changes in cellular function [9,10]. Since then, other extracellular miRNA carriers besides EVs have been identified such as circulating RNA-binding proteins (RBPs), lipoproteins, and nanoparticles [11,12,13,14]. While there has been extensive research on EV-carried miRNAs in interorgan communication, the contribution of these other miRNA vehicles has been barely studied. For this reason, the findings highlighted here refer to miRNAs transported in EVs.

## 2. Classification of EVs

Every eukaryotic cell is able to release EVs to the extracellular environment. They constitute heterogenous populations of membrane particles loaded with specific repertoires of macromolecules including proteins, different RNA subtypes (miRNAs, tRNAs, snoRNAs, snRNAs, and many others), lipids, metabolites, and even organelles such as mitochondria [15,16,17]. From the different subclasses of RNA, miRNAs are among the most abundant cargoes, although the exact contribution is technically challenging to determine due to biases arising from the different profiling methods and from striking divergences in the secretory pattern among cellular models [18,19,20,21]. The aforementioned broad and potent regulatory function of miRNAs has attracted much attention in recent years, to the point where they are now commonly viewed as major effectors of the intercellular communication function mediated by EVs.

EVs can be subcategorized in three main groups based on their size and origin. Exosomes are 50–200 nm in diameter and have the most complex biogenesis, which starts with the formation of endosomal invaginations, thereby creating multivesicular bodies (MVB) (Figure 1). They later fuse to the plasma membrane for the release of exosomes to the extracellular space. This highly intricated and regulated process implies the coordinated action of multiple proteins of the endosomal sorting complex required for transport (ESCRT), Rab GTPases, cytoskeleton, and lipids such as ceramide [22]. In contrast, microvesicles (also named ectosomes) are 100–1000 nm in diameter and derive from direct budding of the plasma membrane, in a process that shares some steps with the formation of exosomes [22] (Figure 1). Apoptotic bodies are large vesicles (up to 5000 nm in diameter) that are released when cells undergo apoptosis [23,24]. This classification is subject to continuous debate as emerging modern technologies, such as asymmetric flow field-flow fractionation (AF4), have recently allowed to distinguish different subclasses of EVs within these major groups such as larger and smaller exosomes [13,24,25]. In addition, non-membranous extracellular nanoparticles, such as exomeres and supermeres (~35 and 28 nm in average, respectively), have also been identified in biofluids and cell supernatants, although their origin and ability to carry miRNAs is still under debate [13,14,26,27] (Figure 1). The repertoire of non-membranous extracellular miRNA carriers also includes RNA-binding proteins (RBPs) and lipoproteins [11,12]. Although further investigation is definitely needed to better define these new elements, available data indicate that each of these novel subcategories of EVs, nanoparticles, RBPs, and lipoproteins are enriched in particular subsets of cargoes and perform distinct biological functions [11,12,13,14,28]. Once released, EVs (and potentially also nanoparticles) can interact with the producer (autocrine communication) or nearby cells (paracrine), or travel through the circulation and interact with distant cells (endocrine). Despite some cases of unspecific interaction/uptake by fusion with the plasma membrane of the recipient cell have been reported, most data support that this is a largely cell-type dependent process, which only occurs if the EV and target cell share the right combination of ligand and receptor in their surfaces [23,29] (Figure 1). This selective interaction can trigger EV internalization into endosomal compartments through various mechanisms. These include endocytosis (either dependent on clathrin, caveolin, or lipid raft-associated flotillins), micropinocytosis, and phagocytosis [23,29]. Once in endosomes, the EV cargo can be recycled, re-secreted, or escape to the cytosol where the miRNAs can be incorporated into the miRNA signaling pathway of the recipient cell, leading to phenotypic changes [23,29,30] (Figure 1).

None of the current methods for EV isolation including ultracentrifugation, size-exclusion chromatography (SEC), or density-gradient centrifugation are able to efficiently separate exosomes from the smallest microvesicles due to their overlapping size [24,25,31]. Furthermore, there are no specific markers for each class of EV, as even classical exosomal markers such as tetraspanins CD9, CD63, and CD81 are also present in microvesicles [23,31,32], thus preventing their selective isolation by using antibody-based immunoaffinity capture. As a consequence of these technical limitations, there are no efficient methodological approaches that provide a reliable isolation and characterization of the individual categories of EVs. For this reason, and although some publications refer to exosomes or microvesicles as the predominant extracellular particle present in their isolates, we will refer to the general term EVs throughout this review.

## 3. Adipose Tissue: Extraordinary Secretory Capacity

Although extracellular miRNAs are potentially involved in the dialog among all organs in our body, they seem to be particularly important for adipose tissue-based communication. This idea is supported by our initial observation that the deletion of Dicer in adipocytes (ADicerKO mice) is associated with a significant downregulation of nearly two-thirds of the circulating EV-carried miRNAs [33]. Similar results were obtained in human immunodeficiency virus-1 (VIH-1)-infected patients who developed lipodystrophy, a common comorbidity associated with this pathogen [34,35]. Adipose tissue undergoing lipodystrophy upon HIV-1 infection shows reduced Dicer expression, especially in the dorsocervical region, through unclear mechanisms [36,37]. Although to a lesser extent than in ADicerKO animals, lipodystrophy HIV-1 patients also display a general downregulation in the expression of many circulating EV-carried miRNAs [34]. These data suggest that a large proportion of circulating EV-carried miRNAs are derived from adipose tissue. Other studies from multiple laboratories have confirmed the extraordinary capacity of adipocytes to secrete EVs, which increases even further in obesity [18,38,39,40,41]. Although there is no experimental evidence in vivo yet, a plausible explanation is that these extra circulating EVs in obese patients and mice may predominantly derive from expanded adipose tissue, although it is important to note that other tissues also respond to obesity by changing their EV secretion rate and cargo content [42,43]. Despite the large body of evidence supporting the extraordinary capacity of adipocytes to secrete EVs, the molecular mechanisms underlying this phenomenon remain unknown. Similarly, the exact molecular pathways responsible for the different miRNA assortments of adipose tissue-derived EVs (AdEVs) in pathological conditions such as obesity have barely been investigated. One such process could be obesity-induced low-grade chronic inflammation, although many other pathways that are dysregulated in obesity could also play a role [44,45].

How miRNAs (and other cargoes) end up in EVs represents a fascinating new area of research. The seminal studies that first observed bioactive miRNAs in EVs already noted that certain subsets of miRNAs were enriched or depleted in these vesicles compared to their parental cells [9,10], suggesting the existence of miRNA selection mechanisms. This observation has been further confirmed by a long list of subsequent reports [46,47,48,49,50,51,52,53,54]. Accumulative research has provided important insights into the molecular mechanisms that determine which miRNAs are sorted in EVs and thus may serve a messenger function. Some tetranucleotide sequences (commonly referred to as EXOmotifs) enriched in the miRNAs that tend to be sorted into vesicles were initially described in some particular cell types [49,51,53], although other motif-independent mechanisms have been also described [48,54]. Our recent multicellular comparative analysis of EV and cellular miRNAs revealed that each cell type including adipocytes uses a particular subset of EXOmotifs to sort miRNAs in EVs [18]. While some EXOmotifs are widely present in the EV-enriched miRNAs released by multiple cell types, others are largely cell-type specific. Interestingly, the miRNAs that are selectively depleted from EVs—and therefore enriched in the cellular body—also contain small sequence motifs that we named CELLmotifs. Similar to EXOmotifs, CELLmotifs can be either broadly distributed across different cell types or cell-type specific. To further complicate the process of miRNA selection or exclusion from EVs, we identified cases of motifs that behave as an EXOmotif in one cell type and as a CELLmotif in another [18]. Strong experimental support for EXOmotifs and CELLmotifs to regulate miRNA distribution comes from the fact that adding or removing these motifs from a given miRNA substantially changes its EV versus cellular destination [18,49,51,53]. The mechanisms guiding the selection of EXOmotif-containing miRNAs or the exclusion of those with CELLmotifs from EVs are far from being understood. Available data suggest that these processes largely rely on RBPs that act selectively in different EXOmotifs and cell types (Figure 1). For instance, in brown adipocytes, ALYREF and FUS participate in the EV loading of CGGGAG-containing miRNAs, the strongest EXOmotif found in this cell type [18]. Other EXOmotif-reader RBPs include SYNCRIP, hnRNPA2/B1, and Lupus La [49,51,52,53]. However, information on the identity of each motif-reader RBP is lacking for the vast majority of identified EXO and CELLmotifs. Similarly, it still remains unclear how the RBP binds to the miRNA and drives it into the forming EV. The fact that these miRNA-binding RBPs are generally not found within the EV cargo [50] point that they may allocate the miRNA in the forming EV while they remain in the cell. Interestingly, recent data have shown that the processes of miRNA sorting as well as the EV release rate are regulated by metabolic factors such as insulin or glucagon [55,56]. This raises the exciting possibility that the dysregulation of miRNA sorting mechanisms may account for the altered EV-miRNA signature and subsequent metabolic abnormalities observed in several metabolic diseases such as obesity and diabetes.

## 4. Adipose Tissue EV Axes

### 4.1. Technical Limitations for EV Study

For adipose tissue, accumulating evidence supports its role as a hub for the release of circulating miRNAs impacting the function of a large number of tissues and cell types. Here, we highlight the most recent findings on AdEV miRNAs that have been shown to shape the metabolism and function of other tissues.

As discussed below, investigation over the past decade has identified numerous miRNAs released by adipocytes and other adipose-resident cells that have an impact on the function of another cell type in a distant organ. However, the study of intercellular communication is a technically challenging area of research: the unequivocal demonstration of the adipose origin of a given miRNA is definitely not trivial, as is the validation of its delivery to a particular target cell type due to several limitations associated with current methodologies (Table 1). Therefore, the use of complementary methodologies are recommended to fully demonstrate an interorgan communication role for a given circulating miRNA. One of the procedures often used to identify a potential target tissue is the exogenous administration of EVs, usually after labelling with luminescent, fluorescent, radioactive, or superparamagnetic tracers [57,58] (Table 1). However, caution must be taken before drawing conclusions from this type of experiment. Exogenously-administered EVs (usually at supraphysiologic concentrations) display a marked tropism for the liver, spleen, lungs, and kidneys, all of which are enriched in phagocytes that seem to perform superior EV uptake [57]. Artificial carriers such as lipid nanoparticles (LNPs) are also preferentially taken up from the circulation by liver or spleen cells, which has favored the development of drug delivery strategies using LNPs to target these tissues [59,60]. In exogenous EV administration experiments, the cellular source for the EVs seems to have a minimal influence in defining the target tissues [57,61]. In contrast, when the EVs are released endogenously, their tissue distribution differs significantly from that of exogenous EVs, with other tissues such as adipose, lung, bone marrow, and gastrointestinal tract leading the EV uptake while liver and spleen were among the tissues with the lowest EV accumulation [57,62]. This suggests that the liver and the other organs where EVs tend to accumulate (lungs, kidneys, and spleen) [57] might act as sink tissues for supraphysiological doses of exogenous EVs and reinforces the need for robust experimental validation to identify interorgan communication axes in vivo (Table 1).

### 4.2. Adipose Tissue to Liver

Intercellular communication between the liver and adipose tissue has been extensively investigated in recent years. Solid accumulative data support that a large number of EV-carried miRNAs reach the liver cells, where they mediate an important regulatory function over hepatic and systemic metabolism (Figure 2). One of the first studies to demonstrate adipose-to-liver communication via extracellular miRNAs was our work with mice deficient for Dicer in adipocytes (ADicerKO) [33]. These mice developed a massive downregulation of circulating EV miRNAs accompanied by elevated hepatic fibroblast growth factor-21 (Fgf21) mRNA and protein levels. Further experiments confirmed that miR-99b released in brown and white adipocyte EVs can target FGF21 in hepatocytes, thereby regulating the hepatic and circulating levels of this key hormone involved in carbohydrate and lipid metabolism [33,63]. Later studies have reported other white adipocyte-derived miRNAs that control systemic glucose metabolism by targeting the liver. This is the case for miR-548ag, whose expression is increased in serum and adipose tissue in obesity. In hepatocytes, this miRNA directly targets the DNA-methyltransferase DNMT3B, which subsequently leads to the upregulation of DPP4, a protein involved in glucose metabolism by degrading incretins such as glucagon-like peptide-1 (GLP-1) [64,65]. Accordingly, the circulating levels of miR-548ag carried by EVs were associated with worsened glucose tolerance and insulin sensitivity [64]. Similar findings have been described for miR-222, which is also elevated in obesity [66,67]. Gonadal WAT was reported to be the major source of EV-carried miR-222, as removal of this fat depot blunted the high-fat diet (HFD)-induced miR-222 increase in the circulation [67]. Mechanistically, miR-222 impairs insulin signaling in hepatocytes through targeting IRS1 [67]. Similarly, miR-4431 is another obesity-associated miRNA that has been linked to worsened glucose tolerance and insulin sensitivity [68]. Although the expression of this miRNA increases several-fold in the WAT and serum of obese individuals, it reaches even higher levels in the liver, indicating that miR-4431 might also be produced directly by hepatocytes [68]. In contrast to previous miRNAs that were upregulated in obesity, miR-141-3p was found to be downregulated in adipose-derived EVs isolated from genetic- and diet-induced obesity models. By targeting phosphatase and tensin homolog (PTEN), this miRNA was shown to inhibit insulin sensitivity and glucose uptake in hepatocytes in vitro (AML12 cells) [69].

Aside from white adipocytes, brown adipocytes are also important senders of AdEV-loaded miRNAs targeting the liver. Indeed, despite their lower mass, brown adipocytes appear to be stronger EV sender cells compared to white adipocytes. This was evidenced by the superior restoration of circulating miRNA levels in ADicerKO mice after the transplantation of wild-type brown adipose tissue (BAT) compared to a lower, but still remarkable, restoration when transplanted white adipose tissue (WAT) [33]. The functional consequences of the BAT-to-liver axis have been further illustrated in cold-exposed mice. In these conditions, brown adipocytes release more EVs, which contain, among others, miR-378a-3p and miR-132-3p. These two miRNAs reach hepatocytes, where they promote gluconeogenesis by targeting p110α (by miR-378a-3p) and repress lipogenesis by SREBF1 targeting (by miR-132-3p) [70,71].

In addition to mature adipocytes, other adipose-resident cells release miRNAs that have important implications for hepatic, and hence, systemic metabolism. A good example is adipose tissue macrophages, whose EV-carried miRNA profile significantly differs in obese versus lean conditions. Some of these miRNAs, such as miR-155 and miR-690, correlate positively or negatively, respectively, with insulin resistance in hepatocytes, adipocytes, and muscle cells [72,73]. Other adipose-resident cells that have been extensively studied in this context are adipose-derived stem cells (ADSCs) due to their major regenerative properties in different tissues and conditions [74], as further outlined in the next sections. Their released EVs can reach the liver, among other tissues, where they exert regenerative functions. For instance, ADSC-released miR-223-3p suppresses hepatic lipid accumulation and fibrosis by E2F1 targeting, while miR-144-3p and miR-486a-3p activate hepatocyte proliferative pathways by suppressing TXNIP expression [75,76].

Reciprocal communication from the liver to adipose tissue mediated by EVs has been much less studied than the reverse direction. In this regard, some reports have shown that hepatic EVs promote glucose uptake in adipocytes through a variety of mechanisms, one of which is the miR-130a-3p-mediated targeting of PHLPP2, a modulator of the AKT-GLUT4 axis [77,78]. Adipocyte targeting of hepatic EVs also plays a role in disease such as obesity and fatty liver. Specifically, obesity implies changes in the miRNA content of EVs released from primary hepatocytes. A subset of these miRNAs, with the strongest effect mediated by let-7e-5p, has been shown to promote lipid accumulation in adipocytes [43]. Some of these miRNAs were found to be elevated in circulating EVs from patients with concomitant obesity and fatty liver but not in those with a fatty liver alone, suggesting a potential involvement of these hepatic miRNAs in fat mass gain [43].

### 4.3. Adipose Tissue to Skeletal Muscle

Overall, the regulatory potential of AdEV-carried miRNAs in skeletal muscle metabolism and function has been less studied than in the liver, but still, a good number of miRNAs have been shown to be involved in this communication axis (Figure 2). One of the first miRNAs reported to be involved in adipocyte-to-muscle communication was miR-130b [79]. This miRNA was found to inhibit the skeletal muscle expression of PGC-1α, a master regulator of lipid oxidation and mitochondrial biogenesis [79]. Later on, miR-27a, whose levels increase in adipose tissue, serum, and skeletal muscle under obese conditions, was shown to promote muscle insulin resistance through the inhibition of PPARγ, at least in an in vitro myotube model [80]. Similarly, obesity-associated circulating miR-222 [66], in addition to hepatocytes, can also reach muscle cells to downregulate IRS1 and insulin signaling, thereby contributing to insulin resistance and glucose intolerance in obesity [67]. Indeed, obesity also alters the secretion and cargo content of EVs secreted by muscle cells. Interestingly, in contrast to the enhanced secretion of EVs by adipose tissue in obesity [40,41], muscle explants from obese animals secrete a lower number of EVs compared to lean explants [42]. The miRNAs contained in these EVs may also be key factors contributing to boost obesity-induced fat gain, as they have been shown to promote lipid accumulation in adipocytes [42]. Beyond the obesity context, there are other situations that illustrate the close reciprocal crosstalk between muscle and adipose tissue mediated by EV-miRNAs. For instance, promoting muscle activity through exercise or synergist ablation results in the enhanced delivery of EV-loaded miR-1 to adipocytes, thereby enhancing catecholamine-induced lipolysis [81,82].

EV-carried miRNAs have also emerged as key elements for the crosstalk between muscle and adipose progenitors. For instance, adipocyte progenitors residing in skeletal muscle (fibroadipogenic progenitors, FAPs) participate in muscle regeneration upon injury. FAPs secrete EV-carried miRNAs (such as miR-127-3p) in a paracrine manner to release muscle stem cells (MuSCs) from quiescence by targeting the myogenic gene S1PR3, thus allowing muscle mass recovery [83,84]. Conversely, MuSCs can also have an influence on adipose progenitors. For instance, MuSCs can reduce the proliferation and differentiation of ADSCs by secreting EV-loaded miR-146-5p, a miRNA previously shown to target insulin receptor [85,86].

### 4.4. Adipose Tissue to Pancreatic Islets

Despite the pivotal role of pancreatic islets in the regulation of systemic metabolism, the study of the potential contribution of adipocyte-released miRNAs to the physiology and pathology of beta cells and other islet cells has barely been investigated in comparison to other organs. However, some recent reports have identified subsets of circulating miRNAs with potential adipocyte origin that regulate beta cell proliferation and activity (Figure 2). For instance, miR-132-3p, whose expression is elevated in visceral adipocytes and pancreatic islets under obese conditions, is linked to enhanced beta cell regeneration [87,88,89,90]. Similarly, miR-15b and miR-146b suppressed insulin secretion in an in vitro model of beta cells [91]. However, whether adipocytes represent the main source of the circulating pool of these three aforementioned miRNAs was not verified [91]. In contrast, the adipocyte origin was confirmed for another cluster of EV-carried miRNAs (miR-29a-3p, miR-200a-3p, miR-218-5p and miR-322-5p) that inhibit early insulin secretion by primary islets and a beta cell line (MIN6) exposed to high glucose conditions [92]. These miRNAs were identified by looking at the EVs released by epidydimal obese adipocytes into cell culture medium [92]. Using an alternative approach, Zhang and colleagues inferred miR-27a-5p as an adipocyte EV-carried miRNA regulating insulin secretion in beta cells [93]. In this case, miR-27a-5p levels increase in mouse and human islets under obesity conditions while its precursor decreases, suggesting an extra-islet origin. The rise in the expression of both mature and precursor miR-27a with obesity specifically in epididymal fat suggested an adipocyte origin for this miRNA. Further in vivo experiments using adipocyte-specific overexpression or neutralization approaches confirmed the EV-mediated transport of this miRNA from adipocytes to islets, where it targeted a calcium transporter (CACNA1C) and thereby blunted insulin secretion [93]. The initial trigger for the elevated secretion of these previous miRNAs in EVs from adipocytes subjected to obesity conditions was not ascertained. In this regard, AdEVs released by adipocytes that were previously exposed to inflammatory cytokines have been shown to promote beta cell apoptosis and impair insulin secretion in recipient beta cells in vitro [44]. These deleterious effects may be mediated, at least in part, by the distinct AdEV-miRNA signature induced by the pretreatment with proinflammatory factors [44]. Taken together, these data suggest that miRNAs released from inflamed adipocytes may be involved in the metabolic deterioration of beta cells in obesity and diabetes. Aside from adipocytes, comprehensive in vivo experiments have also confirmed the release of proinflammatory miR-155-5p in adipose tissue macrophage-derived EVs, which are able to suppress insulin secretion and cellular proliferation in beta cells in vivo [94]. The potential regulatory actions of miRNAs released from adipose tissue cells on other pancreatic islet cells different from beta cells remain to be investigated.

### 4.5. Adipose Tissue to Cardiovascular System

Fat accumulation, especially in visceral depots, is strongly associated with adverse cardiovascular events [95]. The contributing role of adipose tissue in cardiovascular disease (CVD) is illustrated by the protective effect of visceral fat removal in several mouse models of cardiac dysfunction [96,97]. Due to its anatomical proximity to the myocardium, epicardial adipose tissue (EAT), a subtype of visceral fat depot, has received much attention as a potential source of AdEV-carried miRNAs regulating cardiac function. Indeed, several miRNAs released in EAT-derived AdEVs have been shown to regulate metabolism, the production of reactive oxygen species, and contractile function when administered to ventricular myocytes [97,98,99]. These studies suggest important implications for EAT-released miRNAs in the pathogenesis of CVD (Figure 2). Coronary artery disease (CAD) represents the most common form of CVD and is caused by a narrowing of coronary arteries and a subsequent reduction in oxygen supply [100]. Several miRNAs are specifically dysregulated in the EAT (and not in other fat depots such as subcutaneous) of patients with CAD [101,102]. A later study identified several dysregulated miRNAs carried in AdEVs released from the EAT of CAD patients compared to non-CAD patients, many of which were involved in the regulation of cell survival and apoptosis pathways [103]. Although there was no experimental evidence of the delivery of these miRNAs to cardiac cells in vivo, these studies suggest that EV-mediated miRNA delivery likely contributes to CAD-associated myocardial damage and dysfunction. In addition to CAD, AdEVs have been shown to exacerbate other CVD such as atherosclerosis. There is strong evidence that AdEVs can reach and act on endothelial cells [56,104]. In this regard, AdEVs derived from obese visceral fat have been shown to contain high concentrations of miR-132/212 clusters, which promote pro-atherogenic effects by targeting GNA12 and PTEN and ultimately promoting endothelial cell apoptosis and vascular smooth muscle cell (VSMC) proliferation [105].

A factor known to mediate cardioprotective effects is exercise performance. In this regard, the initial research was primarily focused on the role of miRNAs and other secreted factors released by heart-resident cells (cardiomyocytes, endothelial cells, or fibroblast), while the potential contribution of adipocytes has mostly been overlooked [106]. However, recent evidence indicates that BAT-derived miRNAs could also play an important role in exercise-mediated cardioprotection (Figure 2). Upon exercise, brown adipocytes secrete miR-125-5p, miR-128-3p, and miR-30d-5p into their EVs, which prevent cardiac damage induced by a subsequent episode of ischemia/reperfusion due to their suppression of genes involved in apoptosis and the MAPK pathway [107].

The study of factors released by mesenchymal stem cells (MSCs) including ADSCs has gained much interest in recent years as an attractive alternative to cell therapy due to their potent regenerative effects on cardiac muscle [108]. This cardioprotective effect is mediated, at least in part, by extracellular ADSC-derived miRNAs (i.e., miR-146a, miR-205, miR-221, and miR-222) that attenuate cardiac damage in models of ischemia/reperfusion injury by promoting angiogenesis and reducing apoptosis and fibrosis [109,110,111] (Figure 2). The proangiogenic role of ADSC-derived EVs was highlighted by other studies that identified several miRNAs (including let-7, miR-10b, miR-23, miR-24, miR-27, miR-31, miR-486, and miR-1290) associated with enhanced angiogenic responses [112,113,114,115,116]. In addition, miR-196a-5p and miR-425-5p carried by ADSC-derived EVs have been shown to promote cardiomyocyte survival in an in vitro ischemia model [117]. Data also support a protective role of ADSC-derived EVs in ameliorating diabetic retinopathy. Their miRNA cargo is able to reduce oxidative stress, inflammation, and apoptosis while inhibiting abnormal blood vessel growth by modulating key factors such as VEGF, ANG2, and MMP-9, showing promise as a therapeutic agent for early diabetic retinopathy treatment [118]. Taken together, these data indicate that while WAT-derived EV-miRNAs seems to play a deleterious role in the development of cardiovascular disease, those released from BAT and ADSCs exert a protective and regenerative function.

### 4.6. Adipose Tissue to Gastrointestinal (GI) Tract

Adipose tissue and the gastrointestinal tract communicate with each other for the fine-tuned regulation of the uptake, storage, and metabolism of nutrients. For instance, several adipokines such as leptin or adiponectin play a key role in the control of intestinal and gastric functions [119,120]. Conversely, several hormones released by these organs, such as ghrelin, gastric inhibitory polypeptide (GIP), and GLP-1, mediate important functions in adipose tissue metabolism [121,122]. Despite the importance of the adipose tissue–GI tract axis for metabolic control and the fact that EVs from visceral fat show a preferential tropism for the small intestine and colon [123], the role of AdEVs in this communication has been little studied (Figure 2).

Some pioneering studies have addressed the contribution of AdEVs to the most common subtypes of inflammatory bowel disease (i.e., Crohn’s disease and ulcerative colitis). In Crohn’s disease, the fat attached to intestines, known as mesenteric fat, forms an extension named creeping fat (CF), which is unique to this disease [124,125]. CF positively correlates to higher intestinal wall thickness and fibrosis as well as to a lower intestinal lumen diameter, suggesting its participation in the pathogenesis of Crohn’s disease [126,127]. One of the mechanisms by which CF contributes to the development of Crohn’s disease is through the release of EV-miRNAs, which have recently been shown to worsen intestinal fibrosis, a hallmark of Crohn’s disease. Specifically, miR-103-3p carried by EVs released from CF-resident ADSCs promotes fibroblast activation by targeting TGFBR3, which activates SMAD2/3, master regulators of extracellular matrix production [128,129]. In the context of intestinal colitis, proinflammatory miR-155 contained in AdEVs released from obese visceral fat exacerbates colitis in mice [123]. This miRNA is responsible for switching intestinal macrophage polarization toward M1, thereby contributing to elevated intestinal inflammation [123].

Reciprocal communication between the gut and adipose tissue mediated by EVs has also been explored. Most of the attention here has been focused on the EVs released from the bacteria residing in the gut. The potential involvement of the gut microbiome in adipose tissue pathophysiology has largely been suggested based on the strong correlation of the microbiome composition with the degree of adipose tissue accumulation and insulin resistance in obesity [130,131]. Supporting this idea was the initial observation that EVs present in the stool of HFD-fed mice or released from certain microbes enriched in the gut of obese mice could dampen the insulin sensitivity in mice receiving these EVs, both systemically and in adipose tissue [132]. Remarkably, miRNA-like molecules have been identified in bacteria [133]. Intriguingly, these share many common features with miRNAs in eukaryotes, such as similar size and recognition mechanisms, in that they only require partial complementarity to their target sequence [134]. Furthermore, bacterial miRNA-like molecules can associate with the host eukaryotic RISC complex, leading to changes in host gene expression [135]. Recent data point that miRNA-like molecules can be carried by bacterial-derived EVs and delivered to host eukaryotic cells, leading to changes in their cellular function [133,136]. To what extent bacterial EV delivery contributes to adipose tissue biology in health and disease remains to be investigated.

### 4.7. Adipose Tissue to Brain

Despite the lack of a clear explanation of how EVs can cross the endothelial tight junctions comprising the blood–brain barrier [137], several reports have experimentally demonstrated the ability of AdEVs to deliver miRNA cargo to brain cells (Figure 2). In a comprehensive study, Wang and colleagues recently showed that AdEVs display a marked brain tropism (particularly toward the hippocampus) compared to EVs from other sources such as hepatocytes or cells derived from the stromal-vascular fraction (SVF) [138]. The release of AdEV-carried miRNA cargo to brain cells entails pathologic consequences under obese conditions, as a number of miRNAs (miR-9-3p, miR-140-5p, miR-9b-5p, and miR-7a-5p) underlie the cognitive decline commonly associated with obesity. Specifically, miR-9-3p has been shown to regulate neuronal synapse plasticity via BDNF targeting [138].

The aforementioned protective role of miRNAs secreted by ADSCs has also been studied in the context of neuronal protection in ischemic stroke. In this regard, miR-22-3p, miR-26a, and miR-31 reduce neuronal apoptosis and brain injury by targeting the apoptosis-related factors KDM6B, KLF9, and TRAF6, respectively [139,140,141]. In addition to direct effects in neurons, ADSC-derived miRNAs can also reduce brain injury by promoting microglial polarization toward M2 [142,143].

## 5. Clinical Applications of AdEVs

### 5.1. Diagnostics

As extensively illustrated throughout this review, EVs display a unique miRNA signature reflective of specific pathophysiological states. Due to this quality, they have emerged as an excellent source of biomarkers for the early detection and prognosis of a great variety of diseases. By the end of 2023, there were 380 different clinical trials registered on clinicaltrials.gov using EVs for diagnostic purposes [144]. Out of these, the majority (approximately 40%) have focused on cancer, followed by respiratory, nervous, cardiovascular, and endocrine system-related disorders and obesity [144]. With respect to AdEVs, there are several ongoing clinical trials aimed at determining whether AdEVs or vesicles from other sources differ among patient subpopulations whose adipose tissue diverges in activity (e.g., metabolically healthy vs. unhealthy obese; males vs. females) or as a result of metabolic interventions associated with fat mass loss such as exercise or bariatric surgery (Table 2).

The possibility of sampling circulating EVs including AdEVs based on specific surface proteins as a way to obtain critical information about the pathophysiological status of a particular tissue is one of their most promising applications (Figure 3). However, a major limitation here comes from the lack of tissue-specific markers in EVs. Specifically in adipocytes, several EV surface proteins have been previously suggested such as FABP4, perilipin, TGFBI, and adiponectin [38,150,151,152,153,154,155,156]. However, these are not uniquely expressed in adipocytes [157,158,159,160] or may be unspecifically adsorbed to the surface of EVs coming from other cellular origins [153,161], thus representing a major obstacle to achieving a pure isolation of the circulating adipocyte EV subpopulation. The combination of several of these adipocyte-enriched markers may represent an alternative to overcome this limitation [162]. There is still a lack of information on whether adipocyte EVs—as with any other cellular origin—represent a homogeneous population with all vesicles originating from the same subcellular location (e.g., exosomes) displaying the same marker and content or whether there are different subpopulations, each with different markers and potentially different functions. Despite these technical limitations, the enrichment of adipocyte-derived EVs by using the above surface markers and subsequent cargo profiling holds promise for future diagnostics (Figure 3). As an alternative to the still difficult task of capturing enriched adipocyte EVs from the circulation, there is the possibility of sampling adipose tissue and profiling its released EVs ex vivo for diagnostic purposes, a procedure with several inherent limitations that complicate the direct extrapolation to in vivo (Table 1). Among our tissues, adipose is relatively accessible and easy to sample safely and efficiently [163]. In addition, this same method allows for the harvesting of ADSCs. As mentioned throughout this review, ADSCs secrete a particular subset of EV-carried miRNAs, which could also be highly informative for disease diagnosis and prognosis, aside from being an excellent source for therapeutic EVs, as shown below (Figure 3 and next section).

### 5.2. Therapeutics

Due to their extraordinary capacity to deliver cargo to recipient cells, EVs by themselves or as vehicles are being uncovered as therapeutic agents. Certain features of EVs allow them to circumvent some drawbacks associated with existing therapies. In this regard, EVs are less immunogenic and show higher tissue penetration than artificial liposomes [57,164]. In this comparison, EVs achieve an enhanced efficacy for cargo delivery, mainly due to the presence of the CD47–do not eat me signal—on their surface, which prevents their clearance by phagocytes [165]. However, a current limitation for EV-based therapeutics derives from the aforementioned fact that systemic administration of exogeneous EVs leads to certain tissues performing the highest uptake, regardless of the specific cellular source [57,61]. This has prompted the search for certain targeting moieties on the EV surface for enhanced tissue-controlled cargo delivery. For instance, tetraspanin Tspan8 can form a complex with integrin α4 to drive sEV to the pancreas and endothelium [166]. Integrin αvβ5, ApoB1, and albumin have been used to enhance EV delivery to liver cells [161,167,168]. Other examples include the EV allocation of rabies viral glycoprotein (RVG) for brain delivery or sialyl Lewis X (sLeX) and Lewis X glycan ligands for targeting activated endothelial and dendritic cells, respectively [169]. To the best of our knowledge, no targeting moiety has been identified for adipocytes yet.

As outlined throughout this review, an increasing number of publications have demonstrated the pivotal function of AdEVs as mediators of intertissue communication in health and disease. The accessibility of adipose tissue allows for the easy isolation of mature adipocytes or ADSCs, even from the same patient, as a source of EVs for therapeutic purposes (Figure 3). Given the quite divergent AdEV cargo profile between obese and lean patients, the administration of lean-derived AdEVs to obese patients could potentially counteract some of the alterations commonly associated with obesity such as those related to metabolic dysfunction or increased cancer prevalence [92] (Figure 3). Although the use of AdEVs for therapeutic purposes has been marginal to date (only 1% of all registered therapeutic trials using EVs), the use of ADSC-derived EVs is relatively common, being among the top sources for MSC-derived EVs [144]. Due to their exceptional regenerative and immunomodulatory properties, low immunogenicity, feasibility of large-scale production, and safety, MSCs (including ADSCs) are by far the predominant source of EVs for therapeutic purposes (nearly 60% of trials) [144,170,171]. For these reasons, ADSC-derived EVs are emerging as excellent therapeutic agents against multiple disorders such as cardiovascular infarction, wound and ulcer healing, cancer, and many others [172,173,174] (Figure 3, related clinical trials in Table 2).

To date, most EV-based therapeutic trials have used native EVs [144]. However, in addition to transporting naturally loaded miRNAs, EVs hold promise as bioengineered vehicles for user-defined miRNAs and siRNAs. Some of the therapeutic advantages of miRNAs and siRNAs for therapy are that they are easily adaptable to virtually any target (sequences of human genes are known) and their production is cheaper compared to more classical approaches based on small molecules and antibodies, which are costly and can currently target only 0.05% of our genome [175]. Electroporation, sonication, and saponin-mediated permeabilization are some of the techniques used to load EVs with the cargo of interest; however, they have all been shown to promote aggregation and alter their main features [176]. As a result of the recent identification of the mechanisms guiding miRNA sorting in EVs based on EXOmotifs, miRNAs can be bioengineered to generate EVs overloaded with a miRNA of interest as a way to achieve more efficient gene targeting in recipient cells [18,177]. This strategy may open up an exciting new avenue for the future generation of AdEVs carrying a user-defined miRNA repertoire with enhanced therapeutic potential [18] (Figure 3).

## 6. Conclusions

Through numerous studies performed in the last years, it became evident the immense EV-carried miRNA secretory capacity of adipocytes and other adipose resident cells. Circulating EV miRNA cargo has emerged as a key regulatory mediator in intercellular communication, with major implications for physiology and pathology. However, it is very likely that the number and significance of adipose interorgan communication axes through EV-carried miRNAs are just starting to flourish. Hundreds of miRNAs with an adipose origin (as well as other cellular sources) circulate in our biofluids, and we only know the cellular destination and targets of a tiny fraction of them. Many fundamental aspects of this intercellular communication process remain poorly understood. For instance, the molecular mechanisms underlying the extraordinary miRNA secretory capacity of adipocytes in EVs are currently unknown. Furthermore, the biological significance of non-EV circulating miRNA carriers remains to be elucidated. A common question in the field is whether miRNAs secreted by adipocytes (and by any cell type) are packaged in a uniform pool of EVs or in different EV subsets, each addressed to a distinct target tissue and loaded with a restricted miRNA signature and function. The latter could represent a potential explanation for the surprisingly strong biological effects of circulating miRNAs despite the apparently low abundance of individual miRNA species in EVs [178]. Definitely, there are major technical challenges in the field of miRNA-based interorgan communication, some of which have been expressed here. Developing proper tools to assess miRNA origin, destination, and function would allow a full understanding of the predicted vast network of miRNA-based dialog among our tissues and boost very promising therapeutic applications of circulating EV-carried miRNAs for disease diagnosis and treatment.

## Figures and Tables

**Figure 1 ijms-25-13488-f001:**
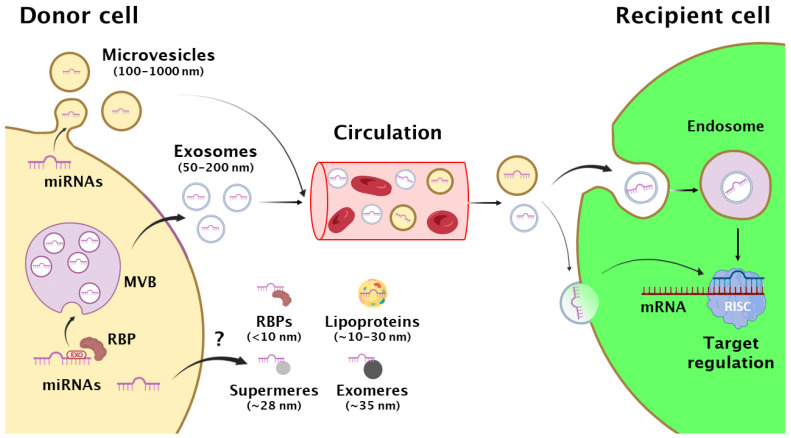
EV-carried miRNAs as new players in intercellular communication. In donor cells, miRNAs are loaded into multivesicular bodies (MVB) for the release of exosomes, or into plasma membrane-derived microvesicles. For exosomes, some short sequences named EXOmotifs promote their sorting into forming exosomes by interacting with RNA-binding proteins (RBP). Exosomes, microvesicles, and apoptotic bodies (not depicted in the figure) are commonly referred as extracellular vesicles (EVs). miRNAs are also secreted through unclear mechanisms in other carriers such as RBPs, lipoproteins, and nanoparticles (exomeres and supermeres). By traveling through the circulation, EVs can reach distant cells, where they are internalized by endocytosis and subsequent incorporation into the recipient’s endosomal system, or to a much lesser extent, through direct membrane fusion. In both cases, miRNAs are able to reach the cytosol of the recipient cell and are incorporated into the miRNA signaling pathway by binding to the RISC complex, ultimately leading to the regulation of the expression of mRNA targets in the recipient cell.

**Figure 2 ijms-25-13488-f002:**
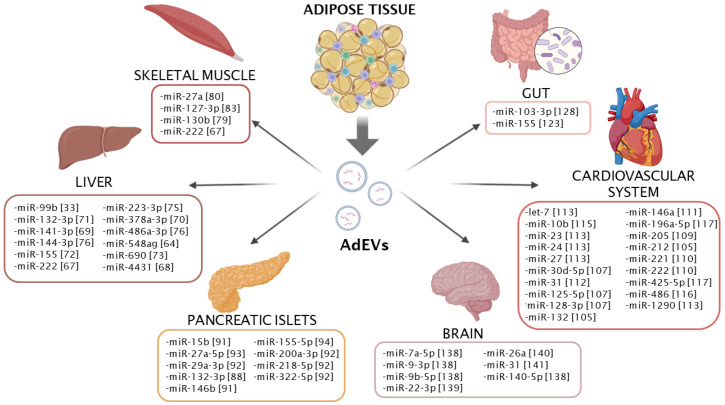
Adipose-derived miRNAs delivered to other metabolic organs. Scheme representing the miRNAs released by WAT/BAT and delivered to the skeletal muscle, liver, pancreatic islets, brain, cardiovascular system, and gut. The numbers in brackets indicate the reference in the main text.

**Figure 3 ijms-25-13488-f003:**
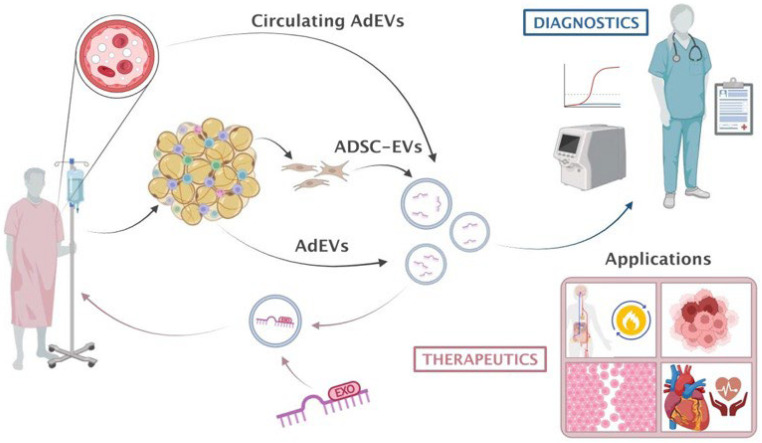
Adipose tissue derived extracellular vesicle applications. AdEVs may derive directly from adipose tissue, adipose-derived stem cells (ADSCs), or sorted from the circulation using enriched surface markers. The content of AdEVs obtained from any of these sources can be used as biomarkers for the diagnosis and prognosis of a great variety of diseases. AdEVs may be also used as therapeutic agents to deliver miRNAs for the treatment of metabolic disorders, tumors, wounds/ulcers, or cardiovascular disease, among other disorders. The incorporation of EXOmotifs might represent a potential new strategy to achieve the selective loading of miRNAs into EVs for enhanced therapeutic potential.

**Table 1 ijms-25-13488-t001:** Current methodologies to study miRNA-based interorgan communication. Table describing the list of common methodologies including their rationality and limitations used to ascribe adipocyte origin to circulating miRNAs as well as for the validation of miRNA delivery to target tissues.

**Procedures to validate adipose origin of a circulating miRNA**		
Method	Rationality	Limitations
In vitro cell lines	miRNAs released to culture medium from adipocyte models 3T3-L1, F442A, PAZ6, etc.	Cell lines might not fully recapitulate miRNA secretion pattern from endogenous adipocytes
In vitro adipocytes derived from ADSCs	miRNAs released to culture medium by adipocytes differentiated in vitro from ADSCs	miRNA secretion pattern might be influenced by differentiation cocktail ingredients and their concentrations, efficiency of differentiation and timing
Obesity	If a circulating miRNA increases in parallel to the increase in fat mass, it might derive from adipocytes	Changes in circulating miRNAs might derive from any of the many cell types affected by obesity
Adipose markers in EV	Capture adipocyte-specific EVs using surface proteins for downstream miRNA profiling	Lack of adipose-specific EV marker, although some are enriched (FABP4, perilipin, TGFBI, Adiponectin)
Adipose tissue explants or primary adipocytes	Incubate fat explants/adipocytes ex vivo and study released miRNAs to the medium	Collagenase digestion might disrupt cell membranes and induce the release of cellular miRNA content
		Incubation in harsh conditions (often serum starved for 24 h) differ from endogenous conditions, potentially inducing cell death and altering miRNA release pattern
		Explants: released miRNAs could derive from any of the cell types.
		Isolated adipocytes: lack of extracellular matrix and cell interactions. Floating conditions can alter released miRNA signature
Viral vector with adipose-specific promoter	An adipocyte promoter (Adiponectin or Fabp4) drives the expression of the miRNA/antagomiR for which the release is being analyzed	Some adipocyte promoters might not be fully specific, e.g., Fabp4 also expressed in macrophages and endothelial cells
Lipectomy/BATectomy or fat transplantation	Removal/transplant of fat from another animal leading to reduced/augmented released levels of a given miRNA	These severe procedures might affect released miRNA signatures from fat and other tissues. Other cell types different from adipocytes might be the predominant miRNA source
Adypocyte-specific miRNA KO, or Adipocyte-Dicer KO (ADicerKO) mice	Deletion of a miRNA gene or Dicer from adipocytes leading to absence/reduction release of that miRNA	Good support for adipocyte origin, but it is costly and time consuming. In animals KO for a miRNA, care must be taken to avoid deleting overlapping genes
		ADicerKO mouse displays a global phenotype, potentially affecting miRNA expression and secretion from other tissues
**Procedures to validate miRNA delivery to a given tissue/cell type**		
Method	Rationality	Limitations
In vitro cell lines	miRNA incorporation upon treatment to cell models in culture	Cell lines might not fully recapitulate miRNA uptake behaviour of endogenous cells
Administration of labelled EV (e.g., by lipophilic dyes) into mice	Detection of the tracer signal in a target tissue	The traffic of exogenous EVs potentially differ from endogenous EVs. Lipophilic dyes can form aggregates similar in size to EVs.
Viral vector with adipose-specific promoter driving the expression of a EV-localized label	Detection of the label signal in another tissue different from adipose	Good support for adipose-to-target tissue EV traffic, though does not proof transfer of a specific miRNA
Viral vector with adipose-specific promoter driving miRNA mimic/antagomiR expression	Detection of enhanced/reduced mature miRNA presence in a tissue different from adipose without changes in precursor miRNA	Good support for adipose-to-target tissue miRNA transfer
Genetic overexpression of a miRNA or its inhibitor in adipose tissue	Detection of enhanced/reduced mature miRNA presence in a tissue different from adipose without changes in precursor miRNA	Good support for adipose-to-target tissue miRNA transfer

**Table 2 ijms-25-13488-t002:** Current clinical trials involving AdEVs and/or EV alterations upon adipose tissue-involved interventions. This table describes current clinical trials retrieved from clinicaltrials.gov that are investigating AdEVs or are performing clinical interventions whose effects are expected to alter circulating AdEVs.

ID	Title	Source EVs	Description
NCT05933707	Effect of Small Extracellular Vesicles From Adipose Tissue on Insulin Action	Subcutaneous WAT and circulating	While in most of the individuals obesity courses with defects in insulin sensitivity, glucose tolerance, dyslipidemia, and hypertension, there is a subpopulation of obese patients ranging from aprox. 4–35%, who do not display these abnormalities [145]. This clinical trial organized by Washington University School of Medicine (USA) aims to investigate whether differences in miRNA and/or lipid content of small EVs could explain the above divergences in insulin sensitivity that is observed between metabolically healthy and unhealthy obese people.
NCT06444646	The Adipose Tissue and Sex-specific Role of Adipose-derived Extracellular Vesicles in Obesity-related Skeletal Muscle Insulin Resistance	Adipocytes isolated from WAT or differentiated from ADSCs, immune cells	This study sponsored by Hasselt University (Belgium) aims to explore potential differences in the EV profile released by adipocytes coming from different depots and the potential communication events that take place between these adipocytes and immune cells. Potential cargo divergences between males an females will also be studied.
NCT06401876	Profiling Extracellular Vesicle Cargo in Obesity and Type 2 Diabetes	Circulating	Bariatric surgery is one of the most effective therapies to reduce obesity and associated comorbidities through unclear mechanisms [146]. This clinical trial sponsored by Mayo Clinic (USA) aims to identify EV molecular signatures responsible for the bariatric surgery-mediated metabolic improvements by profiling circulating EV protein and RNA content in patients before and after this surgical procedure.
NCT06395246	The Usefulness of Circulating Microvesicles (host and Bacterial) in Regulating Metabolic Homeostasis in Obesity Randomized Study of Parallel Arms in Obese Patients Undergoing Caloric Restriction Diet Vs. Early Time-restricted Eating	Circulating, from host and microbiome	Besides the total amount of ingested calories, the timing of their ingestion also matters for obesity development [147]. This is in part mediated by metabolic adaptations in multiple tissues, including adipose tissue and gut microbiome. This clinical trial carried out by Institut Investigacio Sanitaria Pere Virgili (Spain) aims to investigate whether host- and microbiome-derived circulating EVs play a role in the the metabolic profile of patients undergoing full or time-restricted caloric restriction protocols.
NCT05199454	Role of Adiposomes in Diabetes-Associated Endothelial Dysfunction and Restorative Effects of Exercise and Metabolic Surgery	WAT	Vascular dysfunction is one of the hallmarks of obesity and metabolic syndrome [148]. EVs released by adipocytes from obese diabetic patients induce vascular dysfunction, at least in part through their lipid cargo composition [104]. In this clinical trial, investigators from University of Illinois at Chicago (USA) will assess whether clinical interventions that reduce fat mass, such as aerobic exercise or bariatric surgery, can lead to changes in AdEVs content that might be protective for endothelial cells.
NCT04167722	The EXOPRO Study: How Does Prostate Cancer Metastasize? Understanding the Role of Exosomal Communication in Lean vs. Obese Patients	WAT	Obesity is associated to increased risk for cancer development in different tissues, including prostate tumors [149]. Here, investigators from Imperial College London (UK) aim to determine the potential involvement of AdEV miRNAs in prostate cancer proliferation, migration, invasion, apoptosis and epthelial-mesenchymal transition
NCT06253975	A Randomized, Controlled, Multicenter Study of Human Adipose Tissue Derived Extracellular Vesicles Promoting Wound Healing	WAT	This clinical trial hosted by investigators at the Shanghai Ninth People’s Hospital (China) aims to explore potential therapeutic effects of AdEVs in patients with skin ulcers.

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
