# Peer review of "Adipose Tissue as a Major Launch Spot for Circulating Extracellular Vesicle-Carried MicroRNAs Coordinating Tissue and Systemic Metabolism"

_ijms, 2024, doi:10.3390/ijms252413488_

Round 1

Reviewer 1 Report

Comments and Suggestions for Authors

The review article by Diez-Roda, et al. provides an overview of the role of adipose tissue in interorgan communication, focusing on the significance of extracellular vesicle-carried microRNAs. The article emphasizes the importance of circulating miRNAs, particularly those transported by EVs, in metabolic coordination between various tissues. Adipose tissue is highlighted for its ability to secrete a wide range of EV-carried miRNAs into the bloodstream, which act as hormone-like factors. The review also acknowledges the technical limitations in studying the precise origin and destination of these EV-carried miRNAs in living organisms. Despite these challenges, the authors stress the importance of adipose-derived circulating miRNAs in regulating the metabolism and function of other tissues, including the liver, muscle, endocrine pancreas, cardiovascular system, gastrointestinal tract, and brain.

Suggestion for improvement: Line 485-495, the review mentions the lack of tissue-specific markers for EVs, which hinders the precise isolation and study of AdEVs. Interestingly, an article by Thangavel, et al. (iScience 2024) reported a unique technique to isolate and study intact adipocyte-derived EVs from complex tissues and plasma and performed functional analysis in the context of Chagas heart disease. Incorporating and discussing this would strengthen the review by showcasing a promising new avenue for the isolation and study of AdEVs.

Author Response

Comments 1: Suggestion for improvement: Line 485-495, the review mentions the lack of tissue-specific markers for EVs, which hinders the precise isolation and study of AdEVs. Interestingly, an article by Thangavel, et al. (iScience 2024) reported a unique technique to isolate and study intact adipocyte-derived EVs from complex tissues and plasma and performed functional analysis in the context of Chagas heart disease. Incorporating and discussing this would strengthen the review by showcasing a promising new avenue for the isolation and study of AdEVs.

Response 1: we thank the reviewer for her/his suggestion. We have mentioned this method for the isolation of adipocyte-derived EVs from plasma in lines 517-519 of the revised manuscript. 

Reviewer 2 Report

Comments and Suggestions for Authors

Comment on Adipose tissue as a major launch spot for circulating extracellular vesicle-carried microRNAs coordinating tissue and systemic metabolism

The manuscript introduces extracellular vehicles (EVs) containing microRNAs secreted by adipose tissues (AdEVs) and provides a comprehensive explanation of how these microRNA-containing AdEVs influence other tissues and organs. Overall, this work contributes valuable insights into the role of adipose tissue-derived EVs in inter-organ communication and advances our understanding of their biological significance."  

Here are some suggestions for improvement:

1.        It would enhance the manuscript to include a discussion on the mechanism of action of microRNAs in the introduction section.

2.        In section 2, consider introducing different mechanisms of endocytic pathways of EVs, such as clathrin- or caveolae-mediated endocytosis, micropinocytosis, phagocytosis, and lipid-raft-mediated endocytosis.

3.        In section 2, line 66, did you mean “they” instead of “these”?

4.        The title for section 5 might be more suitable as “clinical applications” instead of “therapeutic application”.

Author Response

Comment 1: It would enhance the manuscript to include a discussion on the mechanism of action of microRNAs in the introduction section.

Response 1: We fully agree with the reviewer. Therefore we have performed a detailed description for the molecular pathways by which miRNAs lead to target repression in lines 39-60 of the revised manuscript. 

Comment 2: In section 2, consider introducing different mechanisms of endocytic pathways of EVs, such as clathrin- or caveolae-mediated endocytosis, micropinocytosis, phagocytosis, and lipid-raft-mediated endocytosis.

Response 2: we thank the reviewer for his/her suggestion. We agree that this information was missing. Accordingly, we have modified this part of the review and briefly mentioned the reported mechanisms for EV uptake (lines 108-114 of the revised manuscript). 

Comment 3: In section 2, line 66, did you mean “they” instead of “these”?

Response 3: thank you for pointing this out. We have corrected this typo (line 87 of the revised manuscript).

Comment 4: The title for section 5 might be more suitable as “clinical applications” instead of “therapeutic application”.

Response 4: we thank the reviewer for this suggestion, which is definitely more appropiate. The new tittle can be found in the line 491 of the revised manuscript.